# BTLA: An Emerging Immune Checkpoint Target in Cancer Immunotherapy

**DOI:** 10.3390/ph18121784

**Published:** 2025-11-24

**Authors:** Ming-Cheng Chang, Wan-Chi Lee, Yi-Jou Tai, Ying-Cheng Chiang

**Affiliations:** 1Department of Isotope Application Research, National Atomic Research Institute, No. 1000, Wenhua Rd., Longtan Dist., Taoyuan City 325207, Taiwan; mcchang@nari.org.tw (M.-C.C.); leewc@nari.org.tw (W.-C.L.); 2Department of Obstetrics and Gynecology, College of Medicine, National Taiwan University, Taipei 100226, Taiwan; 3Department of Obstetrics and Gynecology, National Taiwan University Hospital, Taipei 100226, Taiwan; 4Department of Obstetrics and Gynecology, National Taiwan University Hospital Hsin-Chu Branch, Hsinchu City 300195, Taiwan

**Keywords:** BTLA, oncology, B and T lymphocyte attenuator, clinical applications, immune checkpoint, cancer immunotherapy

## Abstract

B and T lymphocyte attenuator (BTLA) is a unique co-inhibitory receptor of the CD28 immunoglobulin superfamily that exhibits dual regulatory functions in immune activation and tolerance. Unlike PD-1 or CTLA-4, BTLA interacts bidirectionally with its ligand HVEM, forming a complex signaling network that shapes immune homeostasis within the tumor microenvironment. Dysregulated BTLA expression has been associated with tumor immune evasion and poor prognosis in several cancers. Owing to its distinctive molecular features and multifaceted immunoregulatory roles, BTLA represents an emerging therapeutic target, particularly in tumors unresponsive to conventional immune checkpoint inhibitors. This review provides a comprehensive overview of BTLA’s structure, signaling mechanisms, and functional implications in tumor immunity and discusses current advances and challenges in BTLA-targeted therapy. Finally, we outline future perspectives on leveraging BTLA modulation to enhance cancer immunotherapy outcomes.

## 1. Introduction

Immune checkpoint molecules are central regulators of immune homeostasis, preserving self-tolerance and preventing autoimmune pathology [1,2]. In cancer, tumor cells exploit these regulatory circuits to escape immune surveillance. The clinical success of antibodies targeting PD-1, PD-L1, and CTLA-4 has revolutionized oncology; however, durable responses remain limited to a subset of patients, and resistance frequently develops [3]. This has prompted the search for additional inhibitory pathways that mediate immune escape and therapeutic resistance.

Among these, B and T lymphocyte attenuator (BTLA) has emerged as a promising next-generation immune checkpoint [4]. By engaging its ligand herpesvirus entry mediator (HVEM), BTLA transmits broad immunosuppressive signals across multiple immune populations, shaping both innate and adaptive immunity. Despite increasing recognition of its role in the tumor microenvironment (TME), the BTLA–HVEM axis remains comparatively underexplored [5,6]. Understanding its biology and therapeutic relevance could broaden the reach of immune checkpoint therapy beyond PD-1 and CTLA-4 blockade (Figure 1).

### 1.1. Overview of Immune Checkpoint Molecules in Cancer Immunotherapy

Research on the most extensively characterized checkpoints—PD-1, PD-L1, and CTLA-4—has demonstrated how inhibitory signaling restrains antitumor immunity. PD-1 on activated T cells binds PD-L1/PD-L2 to suppress proliferation, a mechanism exploited by tumors [7,8]. Antibodies such as pembrolizumab and atezolizumab block this pathway to restore cytotoxic function in cancers including melanoma, non-small-cell lung cancer, and renal cell carcinoma [9,10].

CTLA-4 competes with CD28 for CD80/CD86 binding on antigen-presenting cells, limiting early T cell priming [11]. A study on ipilimumab, the first anti-CTLA-4 antibody, validated checkpoint inhibition but also revealed immune-related toxicities [12]. While these successes confirm the therapeutic value of checkpoint blockades, variable response rates and toxicity profiles highlight the need for novel targets. Consequently, emerging inhibitory receptors such as LAG-3, TIM-3, TIGIT, and BTLA have gained attention as candidates for next-generation therapy [13]. BTLA, a member of the CD28 immunoglobulin superfamily, is unique in its bidirectional interaction with HVEM, a TNF-receptor family member [14]. It is broadly expressed on T, B, dendritic, macrophage, and natural killer cells and signals through ITIM and ITSM motifs recruiting SHP-1/2 phosphatases to dampen activation pathways [14,15].

Within the TME, BTLA often co-localizes with PD-1, TIM-3, and LAG-3 on exhausted T cells, and its upregulation correlates with poor prognosis across multiple malignancies [4,16,17]. Preclinical studies demonstrate that BTLA blockade—alone or combined with PD-1 inhibition—revives effector function and enhances tumor control [18]. However, because the BTLA–HVEM axis intersects with co-stimulatory networks such as LIGHT and CD160, therapeutic design must balance efficacy and immune homeostasis [19,20].

Compared with other immune checkpoints such as PD-1, CTLA-4, TIM-3, and LAG-3, BTLA displays distinctive structural and signaling characteristics. BTLA is a member of the CD28 immunoglobulin superfamily but uniquely interacts bidirectionally with its ligand HVEM, unlike the unidirectional inhibitory signaling observed in PD-1/PD-L1 or CTLA-4/CD80-86 pathways. Structurally, BTLA contains both immunoreceptor tyrosine-based inhibitory motifs and switch motifs in its cytoplasmic tail, enabling it to finely tune immune activation rather than exert a purely suppressive effect. This dual modulatory capacity allows BTLA to maintain immune homeostasis while preventing excessive immune activation. Such structural and functional versatility translates into therapeutic potential, as BTLA blockade could complement existing checkpoint inhibitors and restore anti-tumor immunity in cancers resistant to PD-1 or CTLA-4 targeting therapies.

### 1.2. Brief Introduction to BTLA and Its Role in Immune Regulation

BTLA shares structural similarities with PD-1 and CTLA-4 but engages a distinct ligand, HVEM, forming a bidirectional signaling circuit that can either inhibit or co-stimulate immune responses [4,6]. BTLA is constitutively expressed on multiple immune cell types; upon ligand binding, it transmits inhibitory signals through ITIM/ITSM motifs, recruiting SHP-1 and SHP-2 to suppress receptor signaling, activation, and cytokine release [17,21,22,23,24].

Under physiological conditions, BTLA maintains tolerance and prevents excessive immune activation. In tumors, however, chronic BTLA–HVEM signaling fosters T cell exhaustion and a tolerogenic microenvironment [25]. Blocking or deleting BTLA enhances T cell cytotoxicity and tumor clearance in preclinical models, underscoring its therapeutic potential [26,27]. Because BTLA intersects with other checkpoint and co-stimulatory pathways, deciphering its context-dependent signaling remains essential for rational drug design.

### 1.3. Importance of BTLA in Tumor Immunology and Immunotherapy

BTLA orchestrates immune suppression within the TME by dampening effector functions and sustaining an immunosuppressive milieu. Its expression across CD4^+^/CD8^+^ T cells, B cells, dendritic cells, macrophages, and NK cells enables broad regulation of immune tone [17,28]. Persistent BTLA–HVEM engagement induces T cell exhaustion, limits cytokine secretion and cytotoxicity, and supports regulatory subsets [29,30]. Elevated BTLA expression in tumor-infiltrating lymphocytes correlates with poor prognosis in several cancers, including non-small-cell lung, hepatocellular, melanoma, and gastric carcinomas [31,32].

Therapeutically, BTLA represents a next-generation checkpoint target with potential to enhance existing immunotherapies [4,26]. Dual BTLA and PD-1 blockade synergistically improves tumor rejection in preclinical models [24]. Yet, because the BTLA–HVEM axis integrates with LIGHT and CD160 co-stimulatory signaling, interventions must be precisely tuned to avoid autoimmunity [19,20]. Future work should focus on predictive biomarkers, rational combination regimens, and patient stratification to translate BTLA-targeted therapies effectively.

## 2. Molecular Structure and Signaling Pathway of BTLA

A detailed understanding of BTLA’s molecular architecture and signaling mechanisms is essential to evaluate its role in immune regulation and its potential as a therapeutic target. Unlike more extensively studied checkpoints such as PD-1 or CTLA-4, BTLA engages in a distinctive interaction with HVEM, a member of the TNF receptor superfamily [14]. This pairing forms a non-redundant, bidirectional signaling axis capable of influencing both innate and adaptive immunity. In this section, we examine BTLA’s structural organization and its ligand HVEM, review their expression across immune cell subsets, and outline the intracellular signaling cascades that mediate BTLA’s inhibitory functions. Comparisons with other checkpoint systems highlight BTLA’s unique features and potential advantages in cancer immunotherapy.

### 2.1. Structure of BTLA and Its Ligands HVEM

BTLA is a type I transmembrane glycoprotein in the immunoglobulin (Ig) superfamily. Its extracellular portion contains a single IgV-like domain that binds with high specificity to the cysteine-rich domain 1 (CRD1) of HVEM in a 1:1 stoichiometric ratio [33,34]. This ligand exclusivity distinguishes BTLA from PD-1 and CTLA-4, which interact with multiple partners, and makes the BTLA–HVEM axis structurally stable while functionally non-redundant (Figure 2).

The BTLA cytoplasmic tail contains two conserved motifs, ITIM and ITSM [35]. Upon phosphorylation, these motifs recruit SHP-1 and SHP-2 phosphatases, which dephosphorylate key components downstream of the T cell receptor (TCR) or B cell receptor (BCR) [36,37]. The result is suppression of cell activation, cytokine production, and proliferation. This inhibitory control operates in a context- and cell type-dependent manner, with prominent effects in T and B lymphocytes.

HVEM is composed of four extracellular CRDs [38]. While CRD1 binds BTLA, other domains interact with co-stimulatory molecules such as LIGHT and inhibitory partners like CD160, allowing HVEM to act as a molecular “switch” that integrates opposing immune signals [4]. HVEM can also mediate reverse signaling, transmitting activation or survival cues into HVEM-expressing cells when engaged by BTLA—an ability not shared by PD-1 or CTLA-4 pathways. In the tumor microenvironment, HVEM is often expressed on cancer cells, enabling them to suppress BTLA^+^ immune cells via direct contact, thereby promoting immune evasion and correlating with adverse clinical outcomes [39]. The structural specificity, bidirectional signaling, and integration with other TNFRSF interactions make the BTLA–HVEM axis mechanistically distinct from canonical checkpoint systems [40,41]. These features offer opportunities for precise therapeutic modulation but also present challenges, as blocking BTLA must be balanced against preserving beneficial HVEM-mediated co-stimulation.

### 2.2. BTLA Expression on Immune Cells (B, T, and NK Cells) and BTLA-Mediated Immune Suppression

BTLA is broadly expressed across immune cell subsets, with levels regulated by activation status, inflammatory cues, and the TME (Figure 3). This context-dependent expression enables BTLA to suppress both innate and adaptive immunity through cell-intrinsic and cell-extrinsic mechanisms. In naïve CD4^+^ and CD8^+^ T cells, BTLA ligation inhibits proximal TCR signaling by reducing phosphorylation of ZAP-70 [42] and LAT [43], thereby attenuating calcium flux, MAPK activation, and transcription of effector cytokines (e.g., IFN-γ, IL-2) [27,43]. These effects limit proliferation, cytotoxicity, and memory formation. In chronic stimulation settings, BTLA contributes to the stabilization of exhaustion phenotypes, often in synergy with PD-1 or TIM-3, reinforcing immune suppression in tumors [44].

In B cells, BTLA is constitutively expressed from naïve through memory stages, where its engagement inhibits BCR-mediated activation, proliferation, and antibody production [17,45]. In addition to maintaining peripheral B cell tolerance, BTLA signaling can limit antigen-presenting function [6,27]. Tumor-associated BTLA^+^ B cells may adopt immunoregulatory phenotypes that dampen antitumor immunity.

NK cells are similarly affected, with BTLA signaling curbing the release of perforin, granzyme B, and pro-inflammatory cytokines [46]. This may blunt innate responses critical for early tumor containment. Interestingly, BTLA^+^ NK cells are often enriched in tumor tissues, suggesting that this pathway may be actively co-opted by tumors to silence innate immunity [2,46].

In myeloid cells, BTLA signaling has been implicated in skewing macrophage polarization toward an M2-like, tumor-promoting phenotype [30]. In dendritic cells, BTLA engagement can impair co-stimulatory molecule expression and antigen presentation capacity, fostering a tolerogenic environment that promotes immune escape [32].

A unique aspect of the BTLA-HVEM axis is its bidirectional signaling potential [4]. While BTLA transmits inhibitory signals into immune cells, HVEM may simultaneously receive reverse signals into tumor or stromal cells, influencing their survival and inflammatory response [21]. This dual signaling dynamic complicates therapeutic targeting, as BTLA blockade may have cell-extrinsic effects beyond simply reactivating immune effectors [47]. In the context of cancer, sustained BTLA signaling contributes to the suppression of effector immunity, supports regulatory and tolerogenic immune phenotypes, and facilitates immune evasion. Combined with its expression on multiple immune subsets, BTLA orchestrates a multi-layered suppression network, positioning it as a strategic checkpoint in both adaptive and innate immune arms.

## 3. BTLA in the Tumor Microenvironment

The TME comprises a complex network of malignant cells, immune infiltrates, stromal elements, and soluble mediators, all of which interact dynamically to influence tumor progression and immune responses [48]. Among the many immune-regulatory pathways operating within the TME, BTLA has emerged as a critical checkpoint that shapes the immunosuppressive landscape. Its interaction with HVEM, which is often expressed by tumor cells themselves, enables direct inhibitory signaling at the tumor–immune interface.

In this section, we examine the relevance of BTLA within the TME by first reviewing its expression across different cancer types. We then explore its role in facilitating tumor immune evasion, the crosstalk between BTLA and other immune checkpoints, and the specific effects of BTLA signaling on TILs and Tregs. Understanding these mechanisms is essential for developing BTLA-targeted immunotherapies and identifying optimal combination strategies.

### 3.1. BTLA Expression in Different Tumor Types

Accumulating transcriptomic and immunohistochemical evidence indicates that BTLA is variably expressed across a broad range of human cancers, often in patterns associated with immune suppression or poor prognosis. While BTLA is not typically expressed by tumor cells themselves, its high expression on tumor-infiltrating immune cells—especially dysfunctional T cells and immunosuppressive subsets—suggests a central role in modulating the immune landscape within tumors [17].

In non-small cell lung cancer (NSCLC), BTLA expression is enriched in exhausted CD8^+^ T cells and correlates with reduced effector function and increased tumor burden [49]. In a subset of patients, BTLA^+^ TILs co-express PD-1, LAG-3, or TIM-3, defining a population of deeply dysfunctional T cells less responsive to monotherapy checkpoint blockade [44,50]. Similar co-expression patterns have been observed in hepatocellular carcinoma (HCC), where high BTLA levels were associated with advanced disease stage and a lower density of cytotoxic lymphocytes in the TME [51]. In melanoma, BTLA expression is upregulated following immunotherapy resistance. Analysis of pre- and post-treatment biopsies has shown increased infiltration of BTLA^+^ T cells in tumors refractory to PD-1 blockade, suggesting a possible compensatory role in immune escape [52]. In colorectal cancer, high BTLA expression has been linked to microsatellite-stable (MSS) tumors, which are less responsive to immune checkpoint inhibitors, further supporting its role as a marker of immune exclusion [53].

BTLA expression has also been reported in diffuse large B cell lymphoma (DLBCL) and follicular lymphoma, primarily on intratumoral T cells and B cells, with emerging data indicating that BTLA may predict therapeutic resistance or unfavorable immune subtypes [54]. In these hematologic malignancies, HVEM expression on malignant B cells may engage BTLA^+^ T cells to suppress anti-lymphoma responses [54]. Notably, BTLA mRNA and protein expression levels do not always align, and spatial distribution within the tumor—such as enrichment at the invasive margin—may provide greater insight than bulk expression alone. In epithelial ovarian cancer, detectable BTLA expression in ovarian cancerous tissues was associated with worse disease-free and overall survival for EOC patients [55]. Poor clinical outcome was associated with higher plasma levels of BTLA [56]. The 15-gene, including BTLA, risk signature stratified the EOC cohort which exhibited significant differences in prognosis, gene expression, mutation profiles, immunotherapy response, and drug sensitivity [57]. Emerging multiplex imaging studies are beginning to map BTLA co-expression patterns with other checkpoint molecules and immune cell lineages, offering new diagnostic and prognostic value [58]. The prognostic implications of BTLA expression in various cancer types are listed in Table 1. From public databases, a significant favorable overall survival associated with high BTLA expression is observed in lung cancer, hepatocellular carcinoma, and colorectal cancer (Figure 4). Taken together, BTLA is broadly expressed in tumor-infiltrating immune populations across multiple cancer types. Its enrichment in dysfunctional or suppressive immune cells and its frequent co-expression with other inhibitory receptors make it a promising candidate biomarker and a rational target for combination immunotherapy.

### 3.2. Role of BTLA in Tumor Immune Evasion

Tumor immune evasion is a multifaceted process whereby malignant cells escape detection and destruction by the host immune system. One of the central strategies employed by tumors is the exploitation of immune checkpoint pathways to suppress effector immune cell activity [25]. Within this context, BTLA plays a pivotal role by contributing to the establishment and maintenance of an immunosuppressive TME. BTLA mediates immune evasion through cell-intrinsic and cell-extrinsic mechanisms. On the cell-intrinsic level, its expression on CD8^+^ T cells leads to suppression of TCR signaling, reducing cytokine secretion, cytotoxic granule release, and proliferation [60]. These effects are especially prominent in chronically stimulated, tumor-infiltrating T cells, where BTLA cooperates with other exhaustion markers (e.g., PD-1, TIM-3) to maintain a hyporesponsive state. This allows tumor cells to persist despite the presence of antigen-specific T cells in the TME.

In addition to directly inhibiting cytotoxic T cell activity, BTLA expression on Tregs, B cells, and DCs supports a tolerogenic environment that favors immune escape [21]. BTLA^+^ Tregs may exhibit enhanced suppressive capacity, while BTLA^+^ B cells and DCs tend to exhibit impaired antigen presentation and costimulatory function, thereby limiting the priming and activation of new tumor-specific T cells [72]. A unique feature of BTLA-mediated immune evasion lies in its interaction with HVEM, which is frequently expressed on tumor cells themselves. This creates a direct inhibitory interface between HVEM^+^ tumor cells and BTLA^+^ TILs, allowing tumors to actively downregulate immune activation upon contact [17]. Such cell–cell interactions reinforce localized immune suppression at the tumor front and have been implicated in resistance to immunotherapy.

Moreover, BTLA-HVEM signaling is bidirectional, meaning that engagement of HVEM may also activate survival or pro-growth pathways in the tumor cell, such as NF-κB signaling [29]. This dual benefit—simultaneous suppression of immune attack and enhancement of tumor cell viability—further positions BTLA as a critical immune checkpoint in cancer biology. BTLA also participates in immune exclusion, a phenomenon wherein effector immune cells are physically or functionally prevented from infiltrating tumor cores. High BTLA expression often correlates with immune “cold” phenotypes, characterized by sparse CD8^+^ T cell presence and abundant stromal or suppressive cell populations [73]. BTLA may indirectly promote angiogenesis and lymphangiogenesis through its immunosuppressive effects within the tumor microenvironment. By inhibiting effector T cell activation and enhancing the secretion of vascular growth factors such as VEGF and TGF-β, BTLA signaling supports endothelial proliferation and vessel formation. High BTLA expression in tumors has been correlated with increased metastatic potential. The inhibitory signaling of BTLA dampens antitumor immunity and supports a microenvironment conducive to tumor cell invasion, migration, and distant colonization. These effects may contribute to enhanced tumor vascularization and metastasis. Conversely, BTLA blockade may help restore vascular normalization and improve antitumor immunity. In these settings, BTLA may act in concert with chemokine dysregulation, angiogenic factors, and fibroblast-derived barriers to limit immune accessibility to the tumor.

Finally, tumors may upregulate BTLA indirectly through cytokine-driven mechanisms, including the presence of IL-10, TGF-β, or chronic low-level IFN-γ signaling, which promote the expansion or stabilization of BTLA-expressing immune subsets [27,74]. Taken together, BTLA contributes to tumor immune evasion through multiple, overlapping strategies—by dampening effector responses, reinforcing suppressive immune populations, and enabling direct tumor-to-immune inhibitory signaling. These mechanisms highlight BTLA as a key target in immuno-oncology, particularly in tumors refractory to conventional checkpoint blockade.

### 3.3. Effects of BTLA Signaling on TILs and Tregs

BTLA exerts profound influence on both TILs and Tregs, shaping the immune landscape within the TME [75]. Through its inhibitory signaling, BTLA not only dampens effector T cell function but also reinforces immunosuppressive networks dominated by Tregs, contributing to tumor immune evasion and resistance to immunotherapy [75].

In CD8^+^ TILs, BTLA is frequently co-expressed with other exhaustion markers such as PD-1, TIM-3, and LAG-3. These BTLA^+^ TILs exhibit a dysfunctional phenotype characterized by low expression of cytotoxic molecules (e.g., perforin, granzyme B), reduced production of IFN-γ and TNF-α, and impaired proliferative capacity [44]. Mechanistically, BTLA engagement suppresses TCR-mediated activation through SHP-1/SHP-2 recruitment, leading to downstream inhibition of MAPK and PI3K-Akt signaling pathways [24,76]. This attenuation reinforces T cell exhaustion, particularly in tumors with persistent antigen exposure. BTLA^+^ TILs are often enriched at the invasive margins of solid tumors, where they come into direct contact with HVEM-expressing tumor cells or stromal cells [17,62]. This localized inhibitory interaction further limits effector function precisely where immune attack is needed most.

In CD4^+^ T cells, BTLA signaling similarly reduces helper functions by suppressing cytokine secretion (e.g., IL-2, IL-21) and co-stimulatory molecule expression [27,77]. This compromises their ability to support CD8^+^ T cells and B cells within the TME, contributing to a broader collapse of adaptive immunity. More critically, BTLA appears to play a role in stabilizing and enhancing the suppressive function of Tregs [78]. BTLA is expressed at high levels on tumor-infiltrating Tregs, and its engagement has been associated with upregulation of FOXP3, CD25, and CTLA-4—hallmarks of highly suppressive Treg phenotypes [78]. In murine tumor models, BTLA^+^ Tregs exhibit greater suppressive capacity than their BTLA^−^ counterparts, suppressing CD8^+^ T cell proliferation and cytokine production more effectively [52,77].

The dual role of BTLA—suppressing TIL activation while promoting Treg function—creates a potent immunosuppressive axis within the TME [79]. This is particularly problematic in tumors where both exhausted CD8^+^ T cells and expanded Treg populations coexist, as BTLA signaling reinforces immune dysfunction on both fronts [79]. Figure 5 illustrates the dual role of BTLA in suppressing anti-tumor immunity within the TME. From a therapeutic standpoint, this dual effect positions BTLA as an attractive target. Blocking BTLA may simultaneously restore effector T cell activity and disrupt Treg-mediated suppression, potentially enhancing the efficacy of immune checkpoint blockade [80]. Preclinical studies have shown that BTLA inhibition can improve CD8^+^ T cell function and reduce Treg infiltration or activity within tumors, supporting its inclusion in combination immunotherapy strategies.

## 4. BTLA in Cancer Progression and Prognosis

Beyond its immunomodulatory roles within the TME, BTLA has garnered increasing attention as a molecule linked to cancer progression and clinical outcomes [30]. As high-throughput sequencing and immune profiling technologies advance, BTLA expression patterns have been found to correlate not only with immune cell dysfunction but also with tumor aggressiveness, metastatic potential, and resistance to therapy [6,63,81]. Its expression on multiple immune subsets—combined with the frequent presence of its ligand HVEM on tumor cells—suggests a tightly regulated axis that may influence tumor biology beyond immune suppression alone. This section explores the associations between BTLA and cancer progression, beginning with its relationship to tumor burden, invasiveness, and metastasis. We then examine how BTLA expression levels affect patient prognosis across different cancer types and consider its emerging potential as a prognostic biomarker that may inform risk stratification and treatment decisions.

BTLA may exert immune-regulatory functions in the early phase of tumorigenesis by maintaining immune homeostasis and preventing excessive inflammation. However, as tumors progress, BTLA expression is often upregulated on tumor-infiltrating lymphocytes and tumor cells, contributing to immune exhaustion and facilitating immune escape. In advanced stages, sustained BTLA–HVEM signaling has been correlated with impaired anti-tumor immunity and poor clinical prognosis in several malignancies. These observations suggest that BTLA acts as a context-dependent immune modulator, shifting from a protective to a suppressive role as tumors evolve.

### Correlation Between BTLA Expression and Tumor Progression

Several studies have revealed a strong association between elevated BTLA expression and more aggressive tumor behavior in both solid and hematologic malignancies. This relationship appears to be mediated not only through immune suppression but also via broader influences on the tumor immune ecosystem that indirectly support progression, invasion, and therapeutic resistance.

In NSCLC, BTLA^+^ TILs are enriched in advanced-stage tumors and are particularly abundant at the tumor margin, where interaction with HVEM-expressing tumor cells is most likely. These cells exhibit high expression of exhaustion markers and poor cytotoxic function, correlating with increased tumor size, nodal involvement, and metastatic dissemination [31,52]. High BTLA expression in TILs correlates with shorter overall survival (OS) and progression-free survival (PFS). Retrospective analyses of RNA-seq datasets and immunohistochemical (IHC) studies have consistently shown that patients with BTLA-high immune infiltrates have significantly poorer outcomes, particularly in tumors classified as immunologically “cold” or poorly infiltrated by functional CD8^+^ T cells [82]. Similar findings have been observed in HCC, where elevated BTLA expression was positively associated with vascular invasion, higher histologic grade, and shorter disease-free survival. BTLA expression has been linked to decreased OS and disease-free survival (DFS), independent of tumor size or stage [61,83]. Patients with elevated BTLA levels exhibit lower densities of activated cytotoxic T cells and higher infiltration of suppressive immune subsets, including FOXP3^+^ Tregs [64]. These immunologic features are associated with early recurrence following resection or ablation.

In colorectal cancer, BTLA expression is often elevated in MSS tumors, which tend to exhibit lower immunogenicity and greater resistance to immune checkpoint inhibitors [68]. These tumors also show features of immune exclusion and stromal activation, suggesting that BTLA contributes to the formation of an immune-suppressive, pro-tumorigenic niche that facilitates progression. BTLA levels are more frequently observed in MSS tumors, which tend to be less responsive to immunotherapy [14,27]. In these patients, high BTLA expression correlates with reduced CD8^+^ T cell infiltration, poor response to standard chemotherapy, and lower overall survival rates—further supporting its role as a negative prognostic marker [49].

In melanoma, increased BTLA levels have been linked to disease relapse and poor response to PD-1 blockade [63]. Notably, BTLA expression in pre-treatment biopsies has been proposed as a predictor of early resistance to immunotherapy, as high BTLA co-expression with PD-1 and TIM-3 may reflect a more entrenched exhaustion program within the TIL compartment [84]. Patients whose tumors harbor BTLA^+^PD-1^+^ TILs prior to immune checkpoint therapy have a significantly reduced response rate to PD-1 blockade and worse long-term survival [49,75]. Notably, BTLA expression in baseline biopsies has been proposed as a predictive marker for primary resistance to immunotherapy, with high BTLA levels identifying a subset of patients who may require combination checkpoint inhibition to achieve durable responses [84].

In diffuse large B cell lymphoma (DLBCL) and follicular lymphoma, BTLA expression on T cells and B cells has also been associated with aggressive molecular subtypes and reduced event-free survival [69]. These findings suggest that the BTLA-HVEM axis may contribute to immune escape and unchecked tumor expansion in lymphoid malignancies [32]. Increased BTLA expression on tumor-infiltrating T cells has been associated with lower event-free survival and treatment resistance. High BTLA expression also correlates with elevated expression of other exhaustion markers (e.g., LAG-3, TIGIT), suggesting deeper immune dysfunction in these disease settings [70,85].

Mechanistically, the BTLA-HVEM interaction may reinforce tumor progression through multiple channels, including by limiting effector T cell-mediated cytotoxicity, enhancing Treg stability and function, and potentially engaging reverse signaling in HVEM-expressing tumor cells to promote their survival or resistance to apoptosis [32,86,87]. Importantly, BTLA expression often adds independent prognostic value even when accounting for conventional clinical variables such as tumor stage, grade, or mutational status [82,88]. Its integration into multi-parameter prognostic models or immune risk scores may improve precision in stratifying patients for therapy selection and outcome prediction. BTLA-mediated immune regulation beyond T cells in the TME is shown in Figure 5. Collectively, these findings indicate that BTLA is not only a suppressor of anti-tumor immunity but also a marker of poor clinical prognosis across diverse malignancies. As BTLA-targeting therapies move toward clinical evaluation, BTLA’s expression profile may serve as both a predictive and prognostic biomarker, guiding the design of tailored immunotherapy regimens [64,65].

Taken together, these observations highlight a consistent theme across cancer types: high BTLA expression is not merely a marker of immune suppression but also a feature linked to biologically aggressive tumors. These insights underscore the importance of further investigating BTLA as a functional contributor to tumor progression and as a possible therapeutic target to disrupt tumor-promoting immune pathways.

## 5. BTLA as a Target for Immunotherapy

The advent of ICIs has revolutionized cancer treatment, particularly with the success of agents targeting PD-1, PD-L1, and CTLA-4. However, despite significant breakthroughs, a substantial proportion of patients remain non-responsive or develop resistance to existing immunotherapies. This clinical limitation has fueled efforts to explore additional checkpoint molecules, including BTLA, as potential therapeutic targets (Table 2).

BTLA’s unique biology—broad expression across immune subsets, engagement with HVEM, and involvement in both innate and adaptive suppression—makes it an attractive but complex candidate for immune modulation. In contrast to PD-1 or CTLA-4, BTLA forms a bidirectional signaling axis with its ligand, influencing not only T cells but also tumor cells and APCs [97]. These characteristics suggest that BTLA blockade may restore antitumor immunity through mechanisms distinct from current ICIs and possibly enhance the efficacy of existing checkpoint regimens when used in combination. In this section, we provide an overview of current BTLA-targeting ICIs, review key preclinical and clinical studies, discuss combination strategies with other ICIs, and address emerging concerns regarding safety and immune-related toxicities.

### 5.1. Overview of Current ICI Targeting BTLA

Although BTLA is a relatively novel immune checkpoint target, multiple investigational therapies have been developed to block its inhibitory effects and reinvigorate antitumor immunity. These agents are primarily monoclonal antibodies designed to disrupt the BTLA-HVEM interaction, though other therapeutic formats, such as bispecific antibodies and fusion proteins, are under exploration (Table 3).

One of the most advanced BTLA-targeting agents in clinical development is tifcemalimab (JS004), a fully human IgG4 monoclonal antibody targeting BTLA, which was previously referred to as TAB004 during preclinical development. Early clinical studies have demonstrated encouraging activity across multiple tumor types. In patients with previously treated advanced lung cancer, tifcemalimab combined with the PD-1 antibody toripalimab produced manageable safety and preliminary efficacy, supporting the rationale for dual checkpoint blockade of the BTLA/HVEM and PD-1/PD-L1 pathways [84]. Similarly, a Phase I trial in relapsed or refractory lymphoma showed that tifcemalimab, either as monotherapy or in combination with toripalimab, achieved durable responses and immune activation with an acceptable toxicity profile [85]. Currently, the major therapeutic agents under development targeting the BTLA–HVEM axis are summarized in Table 2.

In addition, dual blockade of BTLA and other immune checkpoints (such as PD-1 or CTLA-4) has shown synergistic antitumor effects in preclinical models [75]. These molecules aim to streamline combinatorial immune activation while potentially reducing toxicity by limiting systemic exposure. Beyond monoclonal antibodies, the design of CAR-T cells incorporating optimized co-stimulatory TNFRSF domains (e.g., HVEM, OX40, 4-1BB) is being explored to enhance functionality in adoptive cell therapy [40]. Early data suggest that T cells lacking BTLA show enhanced persistence and cytotoxic function in solid tumor models. Overall, while BTLA-targeting agents remain in early-phase development, their design reflects an appreciation for the receptor’s unique signaling context and non-redundant role in immune regulation. As more agents enter clinical evaluation, comparative studies will be needed to determine the optimal modality, dosage, and treatment setting for BTLA-directed immunotherapy.

### 5.2. Advances in BTLA Blockade and Combination Immunotherapy

BTLA has emerged as a promising next-generation immune checkpoint. Preclinical studies demonstrate that BTLA blockade restores T cell effector function, increases IFN-γ production, and enhances tumor control in melanoma, colon, and hepatocellular carcinoma models [5,17,68]. Mechanistically, it reduces regulatory T cell and tolerogenic dendritic cell activity, decreases FOXP3 and IL-10 expression, and reprograms the tumor microenvironment toward immune activation [21,68]. In contrast, ovarian cancer—with low immune infiltration and dominant immunosuppressive networks—may respond less to BTLA monotherapy, underscoring the need for combinatorial or multimodal strategies [87,88,89,90,91]. Clinically, several trials are evaluating the anti-BTLA antibody tifcemalimab (JS004/icatolimab), alone or in combination with the PD-1 inhibitor toripalimab, across malignancies such as Hodgkin lymphoma, NSCLC, SCLC, and ESCC (Table 3). Combination approaches are gaining momentum, as dual BTLA/PD-1 blockade yields synergistic antitumor activity and reduced Tregs in preclinical models [60,93], while pairing BTLA inhibition with CTLA-4 or TIM-3 blockade may overcome non-redundant inhibitory pathways [94]. Collectively, preclinical and early clinical data suggest that BTLA inhibition, particularly when combined with other immune checkpoint inhibitors or conventional therapy, may reinvigorate exhausted T cells and help overcome resistance to PD-1 blockade.

### 5.3. Potential Side Effects and Toxicity Concerns with BTLA Blockade

ICIs are associated with immune-related adverse events (irAEs), arising when the immune system attacks normal tissues [103]. Typical irAEs include dermatitis, colitis, pneumonitis, hepatitis, and endocrine dysfunctions, which are well-documented side effects for PD-1 and CTLA-4 agents. Given BTLA’s role as a co-inhibitory receptor, its blockade may similarly unleash autoimmunity and systemic inflammation. The potential side effect profile for BTLA blockade is inferred from its mechanistic similarity to other ICIs. By removing BTLA-mediated suppression, elevated T cell activation may lead to off-target tissue damage, particularly in organs susceptible to autoimmunity such as the skin, gut, liver, lungs, and endocrine glands [104,105]. While clinical data specific to BTLA inhibitors remain limited, early preclinical reports and conceptual insights suggest that irAEs—including colitis, dermatitis, pneumonitis, hepatitis, and endocrinopathies like thyroid dysfunction—could emerge similarly [26,106]. Moreover, the bidirectional signaling nature of BTLA-HVEM introduces additional complexity. Blocking BTLA may not only hyper-activate immune cells but also disrupt HVEM-mediated survival signaling in non-immune tissues [27,32]. This could theoretically alter tissue homeostasis or provoke inflammatory cascades in tumor or stromal compartments, intensifying tissue-specific toxicity.

Combination regimens involving BTLA and other ICIs—such as PD-1 or CTLA-4 inhibitors—may elevate the risk and severity of irAEs. Historically, combinations like PD-1 + CTLA-4 blockade have significantly increased Grade 3–4 toxicities (10–30% or higher) compared to monotherapy [107]. Adding BTLA blockade may further exacerbate immune overshoot, necessitating careful dose optimization and safety monitoring. Preclinical models suggest potential mitigation strategies, including dose titration, intermittent dosing, or local rather than systemic delivery to minimize systemic autoimmunity [84]. Biomarker-guided patient selection, such as baseline autoantibody screening or cytokine profiling, may also help identify individuals at higher risk of irAEs.

In clinical application, surprise-onset toxicities like immune-related myocarditis or neurologic events—though infrequent—must be considered when new immunomodulatory agents like anti-BTLA are deployed. In summary, while BTLA blockade holds promise for reactivating suppressed antitumor immunity, the potential for irAEs mirrored in PD-1 and CTLA-4 inhibitors underscores the need for meticulous clinical monitoring. Trials should include irAE surveillance protocols, early intervention algorithms (e.g., corticosteroids), and predefined criteria for treatment modification to safely harness the therapeutic benefit of BTLA-targeted immunotherapy.

## 6. BTLA and Tumor Immunity: Beyond T Cells

While most research on BTLA has focused on its inhibitory effects on T cell activation, accumulating evidence suggests that BTLA also plays pivotal roles in regulating other immune cell subsets, including B cells, NK cells, and certain myeloid populations [21]. These interactions expand the immunoregulatory scope of BTLA far beyond the T cell compartment, influencing both adaptive and innate immune responses in cancer. Given the complexity of the TME, where diverse immune cell types contribute to tumor control or progression, understanding BTLA’s functions outside of T cells is crucial for developing more comprehensive therapeutic strategies. In particular, BTLA’s involvement in modulating B cell antibody production, NK cell cytotoxicity, and the activity of dendritic cells or macrophages may have profound implications for antitumor immunity, shaping both humoral and cell-mediated immune landscapes.

### 6.1. BTLA’s Role in Modulating B Cell Responses in Cancer

BTLA is constitutively expressed on the majority of peripheral B cells, with especially high levels on naïve and germinal center B cells. Its engagement with HVEM delivers inhibitory signals that suppress BCR-mediated activation, proliferation, and antibody secretion. Mechanistically, BTLA signaling in B cells recruits SHP-1/SHP-2 phosphatases, leading to dephosphorylation of key proximal BCR signaling molecules such as Syk and BLNK [108,109]. This signaling cascade reduces downstream activation of NF-κB and MAPK pathways, ultimately dampening B cell-driven immune responses. In the context of cancer, BTLA’s regulation of B cells can have dual implications. On the one hand, suppression of B cell activity may impair antitumor humoral immunity, reducing the production of tumor-specific antibodies that could facilitate antibody-dependent cellular cytotoxicity (ADCC) by NK cells or promote antigen presentation [66]. On the other hand, BTLA-mediated control of B cells may limit the protumorigenic functions of certain B cell subsets, such as regulatory B cells (Bregs), which can produce IL-10 and TGF-β to suppress effector T cell and NK cell activity [110,111]. Clinical studies have shown that elevated BTLA expression on circulating or tumor-infiltrating B cells correlates with poor prognosis in several cancers, including hepatocellular carcinoma and non-small cell lung cancer [4]. These findings suggest that BTLA may skew B cell responses toward an immunosuppressive phenotype in the TME. Experimental BTLA blockade in B cells, either via genetic deletion or monoclonal antibodies, has been reported to enhance antigen-specific antibody responses and increase the ability of B cells to prime CD4^+^ T cells in preclinical tumor models [27,112]. Given the complex interplay between B cells and other immune populations in cancer, targeting BTLA in B cells may provide a two-fold therapeutic benefit: restoring tumor-specific humoral immunity and reducing Breg-mediated immune suppression. However, translating these findings into clinical strategies will require a deeper understanding of how BTLA expression on different B cell subsets varies across tumor types and disease stages, as well as the impact of BTLA inhibition on overall immune homeostasis.

### 6.2. Impact on NK Cells and Other Immune Cell Subsets

BTLA expression on NK cells has been increasingly recognized as an important regulatory mechanism influencing innate immunity in the TME. Although NK cells are traditionally viewed as frontline effectors against tumor cells through direct cytotoxicity and cytokine secretion, BTLA engagement with HVEM can attenuate their effector functions. Mechanistically, BTLA signaling in NK cells inhibits the phosphorylation of key activating receptors such as NKG2D and NKp30, leading to reduced release of perforin and granzyme B [46,113]. Additionally, BTLA-mediated suppression limits NK cell production of IFN-γ, a critical cytokine for orchestrating both innate and adaptive antitumor responses [114]. In several cancer models, including hepatocellular carcinoma and gastric cancer, tumor-infiltrating NK cells exhibit elevated BTLA expression compared to peripheral NK cells from healthy controls [14,17]. This upregulation is often associated with a more exhausted NK cell phenotype, characterized by diminished cytotoxic activity and reduced expression of activating receptors. Blocking BTLA–HVEM interactions in preclinical settings has been shown to partially restore NK cell cytotoxicity and increase tumor clearance, suggesting that BTLA may be a key checkpoint in restraining NK cell-mediated immunity [46,71]. Beyond NK cells, BTLA is also expressed on other immune subsets relevant to tumor immunity. For example, DCs expressing BTLA can adopt a tolerogenic phenotype, producing less IL-12 and more IL-10, thereby skewing T cell responses toward a regulatory profile [72,115]. Recent studies have shown that BTLA is expressed on macrophages, where it functions as an inhibitory checkpoint influencing the balance between M1 and M2 polarization. Activation of BTLA can recruit SHP-1/2 phosphatases and suppress NF-κB-mediated pro-inflammatory signaling, thereby promoting a shift toward an M2-like, anti-inflammatory phenotype. This BTLA-mediated transition may facilitate tumor-associated macrophage reprogramming, contributing to immune tolerance and tumor progression. Thus, targeting BTLA in combination with macrophage-modulating strategies may provide synergistic therapeutic potential in reshaping the tumor immune microenvironment [116,117]. Similarly, myeloid-derived suppressor cells (MDSCs) may leverage BTLA-mediated pathways to enhance their suppressive capacity, further contributing to immune evasion. The collective effect of BTLA expression across these innate and myeloid compartments is the reinforcement of an immunosuppressive TME that impairs both direct tumor cell killing and the priming of effective adaptive responses. Importantly, the relative contribution of NK cells versus other BTLA-expressing innate cells to tumor progression likely varies across cancer types and disease stages. Therapeutically, targeting BTLA in NK cells and other innate immune cells could complement T cell–focused immunotherapies, potentially overcoming resistance in tumors with low T cell infiltration (“cold” tumors) [117,118]. However, given BTLA’s role in maintaining immune homeostasis, systemic blockade may carry risks of hyperinflammation or autoimmunity, highlighting the need for selective targeting strategies—such as bispecific antibodies that preferentially block BTLA in the TME or cell-specific delivery systems [119,120].

## 7. Challenges and Future Directions in BTLA Research

Despite significant advances in understanding the biology of B and T lymphocyte attenuator (BTLA) and its role in tumor immunology, the translation of BTLA-targeted strategies into clinical success remains in its early stages. While preclinical studies have highlighted its potential as a novel immune checkpoint for cancer therapy, several challenges continue to hinder its full therapeutic exploitation. Current BTLA-directed interventions face limitations in efficacy, patient selection, and safety, partly due to the heterogeneous expression of BTLA across tumor types and immune cell subsets, as well as the complexity of its bidirectional signaling with HVEM [4,84]. Furthermore, the absence of validated predictive biomarkers makes it difficult to identify patients who are most likely to benefit, increasing the risk of underwhelming clinical trial outcomes. The key challenges and future directions in BTLA-targeted immunotherapy are shown in Figure 6. On the drug development front, the field still requires innovation in designing potent, selective BTLA inhibitors or modulators that can be integrated into combination regimens without causing excessive immune-related toxicity. Finally, as oncology moves toward precision medicine, the future of BTLA immunotherapy will depend on tailoring treatment to individual tumor and immune profiles, leveraging multi-omics data, and developing adaptive clinical trial designs. This section examines these critical challenges and outlines future directions to unlock the full potential of BTLA-targeted interventions.

### 7.1. Current Limitations and Biomarker Needs for BTLA-Targeted Immunotherapy

Although BTLA has emerged as a promising immune checkpoint target, its clinical translation remains constrained by several limitations. One major challenge is the variability of BTLA expression across tumor types, immune cell subsets, and even among patients with the same cancer. This heterogeneity complicates the design of broadly effective therapies and may partially explain the modest efficacy observed in early-phase trials. In addition, the dual role of BTLA—acting as an inhibitory receptor on effector cells while potentially delivering co-stimulatory signals via HVEM—creates complexity in predicting therapeutic outcomes. Overactivation of the immune system through BTLA blockade can also raise safety concerns, including autoimmune manifestations and immune-related adverse events, particularly when combined with other checkpoint inhibitors [121,122]. Another critical limitation is the absence of validated predictive biomarkers to guide patient selection. Without robust biomarkers, trials risk enrolling patients unlikely to respond, diluting efficacy signals and hindering drug approval. Potential candidates under investigation include BTLA expression levels on TILs, HVEM expression on tumor or stromal cells, and transcriptional signatures reflecting T cell exhaustion phenotypes. Circulating immune profiling and multiplex immunohistochemistry may further refine patient stratification. Moreover, functional biomarkers such as changes in IFN-γ production or reduction in regulatory T cell activity after ex vivo BTLA blockade could serve as pharmacodynamic indicators. The development of such biomarkers will be essential to optimize dosing, minimize toxicity, and increase the probability of clinical benefit.

### 7.2. Innovation in BTLA Modulation and Prospects for Personalized Medicine

Future progress in BTLA-targeted therapy will depend heavily on innovation in drug design and integration with precision oncology. Beyond conventional monoclonal antibodies, novel formats such as bispecific antibodies targeting BTLA and PD-1, Fc-engineered antibodies with enhanced receptor engagement, and fusion proteins that modulate BTLA-HVEM interactions are under exploration [4]. Small-molecule inhibitors capable of disrupting BTLA signaling at the intracellular domain level may offer oral delivery options and improved pharmacokinetics [123,124]. Conversely, in certain pathological contexts where BTLA exerts protective roles, BTLA agonists could be developed to enhance immune tolerance, such as in transplantation or autoimmune disorders—highlighting the versatility of BTLA modulation. Personalized medicine approaches will be critical in realizing the full potential of BTLA-targeted strategies. Integration of genomic, transcriptomic, and proteomic profiling can identify patients whose tumor immune landscapes are particularly dependent on BTLA-mediated suppression. Machine learning algorithms could analyze multi-omics datasets to generate predictive scores for BTLA therapy responsiveness. Additionally, adaptive clinical trial designs—where patient enrollment and treatment arms are dynamically adjusted based on interim biomarker analyses—could accelerate the identification of optimal drug-biomarker combinations. Ultimately, successful BTLA-targeted immunotherapy will likely require a tailored approach, combining BTLA modulation with other ICIs, targeted therapies, or immunomodulators based on each patient’s tumor biology and immune status. This precision-driven paradigm holds promise for transforming BTLA from a niche checkpoint into a clinically impactful target in cancer immunotherapy.

## 8. Conclusions

BTLA has emerged as a critical immune checkpoint in tumor immunology, functioning as a key regulator of immune homeostasis within the tumor microenvironment. Through its interaction with HVEM, BTLA delivers potent inhibitory signals that modulate T cell, B cell, NK cell, and other immune subset activities, thereby contributing to immune evasion and tumor progression. Its broad expression across immune cell types and its presence in multiple malignancies underscore its relevance as both a mechanistic driver of tumor immune suppression and a potential prognostic biomarker. The growing understanding of BTLA’s biology offers promising avenues for the development of novel cancer immunotherapies. Targeting BTLA may restore effector immune cell function, reprogram the tumor microenvironment, and enhance anti-tumor responses, particularly when combined with other immune checkpoint inhibitors such as PD-1 or CTLA-4. However, clinical translation will require addressing several challenges, including the heterogeneity of BTLA expression, its dual regulatory roles, and the identification of reliable predictive biomarkers to guide patient selection.

Despite encouraging preclinical findings, notable translational gaps persist in the development of BTLA-targeted therapies. Current preclinical models, including murine tumor systems, fail to fully replicate the complexity and heterogeneity of the human tumor microenvironment, limiting their ability to predict clinical efficacy. Furthermore, the design of clinical trials incorporating BTLA inhibitors remains challenging due to the lack of validated biomarkers for patient selection, uncertainties regarding optimal combination strategies with existing immune checkpoint inhibitors, and the potential for immune-related adverse effects. Bridging these gaps through improved humanized models, biomarker-driven trial design, and integrated translational research will be crucial for realizing the therapeutic potential of BTLA in clinical oncology.

Looking forward, the integration of BTLA-targeted strategies into the broader immuno-oncology landscape holds significant potential. Combined with precision medicine approaches leveraging multi-omics profiling and AI-based predictive tools, advances in drug design—ranging from monoclonal antibodies to bispecific constructs and small-molecule modulators—could enable more personalized and effective interventions. While current clinical data are still limited, ongoing research is steadily moving BTLA from a promising preclinical target toward viable clinical application. With continued scientific and translational progress, BTLA-targeted immunotherapy may soon become an important addition to the expanding arsenal of cancer treatment options.

## Figures and Tables

**Figure 1 pharmaceuticals-18-01784-f001:**
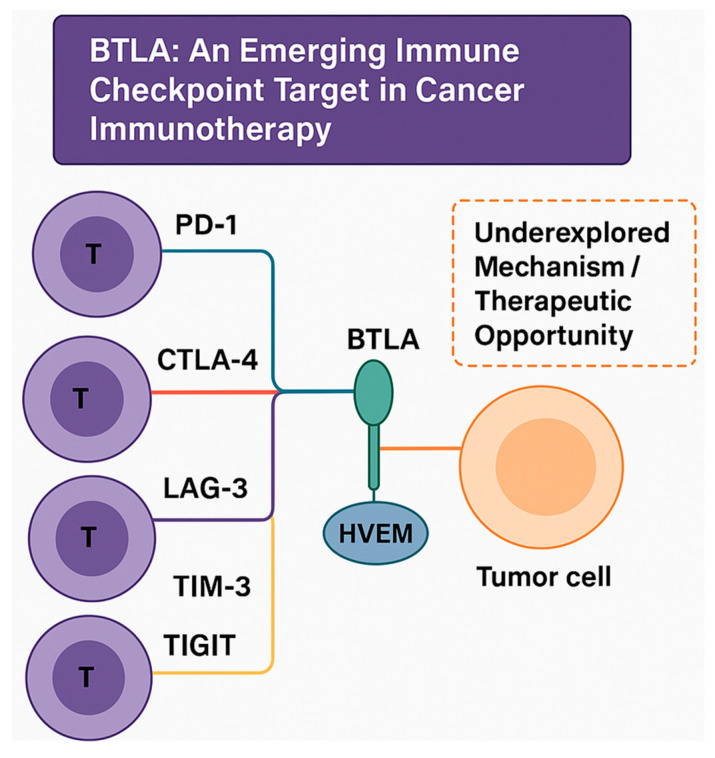
Conceptual overview of immune checkpoint pathways highlighting BTLA as an emerging therapeutic target. This illustration presents established inhibitory checkpoints—PD-1/PD-L1, CTLA-4/CD80-86, LAG-3, TIM-3, and TIGIT—on T cells that interact with tumor or antigen-presenting cells. BTLA is emphasized as a distinct member of the CD28 superfamily that engages HVEM in a bidirectional signaling manner. This BTLA–HVEM axis contributes to tumor immune evasion yet remains an underexplored mechanism and therapeutic opportunity, underscoring the rationale for deeper investigation into its biology and translational potential in cancer immunotherapy.

**Figure 2 pharmaceuticals-18-01784-f002:**
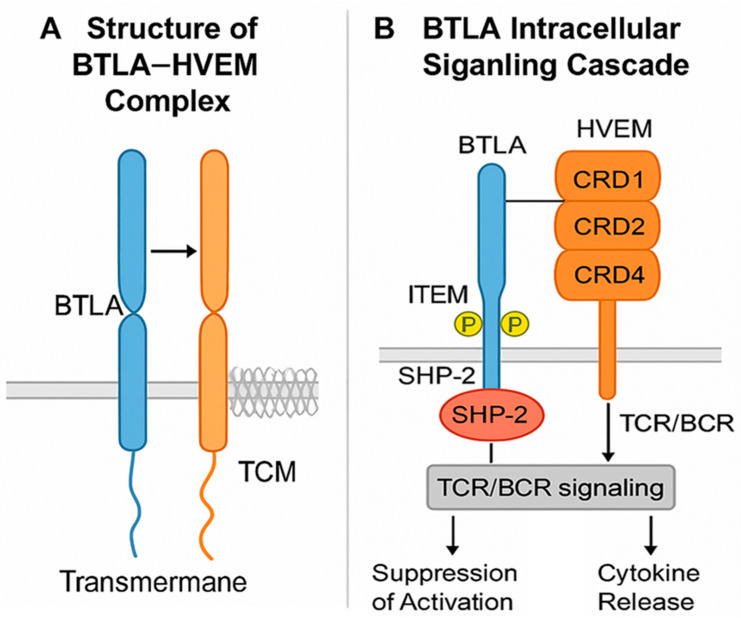
Molecular structure and signaling pathway of BTLA. (**A**) Structure of the BTLA–HVEM complex: BTLA (B and T lymphocyte attenuator) is a type I transmembrane glycoprotein belonging to the immunoglobulin superfamily. Its extracellular IgV-like domain binds specifically to the cysteine-rich domain 1 (CRD1) of HVEM (herpesvirus entry mediator), a TNF receptor superfamily member. This 1:1 interaction forms a structurally stable but functionally non-redundant inhibitory complex. (**B**) BTLA intracellular signaling cascade: Following engagement with HVEM, the cytoplasmic tail of BTLA becomes phosphorylated at conserved ITIM and ITSM motifs, which recruit SHP-1 and SHP-2 phosphatases. These phosphatases dephosphorylate downstream signaling molecules of the T cell receptor (TCR) or B cell receptor (BCR), resulting in suppression of activation, reduced cytokine production, and attenuation of proliferation. HVEM, containing four CRDs (CRD1–CRD4), can also interact with other ligands such as LIGHT or CD160, acting as a bidirectional signaling hub that integrates both stimulatory and inhibitory cues. (Note: BTLA: blue color; HVEM: orange color).

**Figure 3 pharmaceuticals-18-01784-f003:**
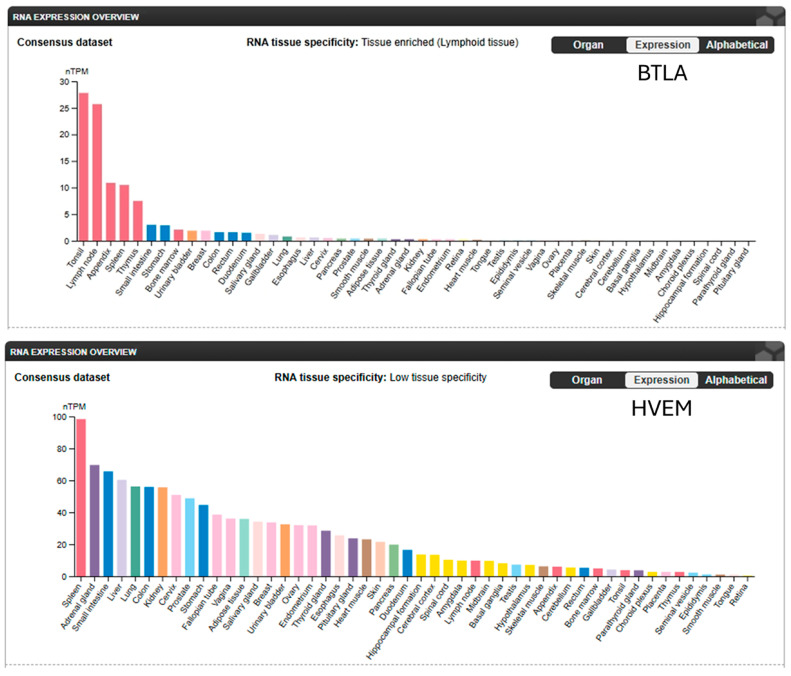
BTLA and HVEM expression levels in normal human tissues. BTLA expression is highly tissue-specific, mainly in lymphoid tissue. HVEM expression demonstrates low tissue-specificity. (Adapted from The Human Protein Atlas: https://www.proteinatlas.org/ENSG00000186265-BTLA/tissue; https://www.proteinatlas.org/ENSG00000157873-TNFRSF14/tissue (both accessed on 1 November 2025).

**Figure 4 pharmaceuticals-18-01784-f004:**
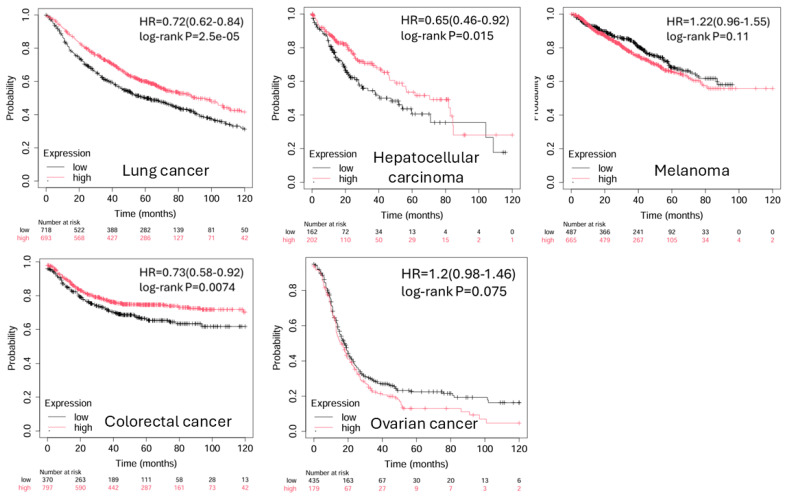
Kaplan–Meier plots of BTLA expression levels in cancers. The significant favorable overall survival associated with high BTLA expression is observed in lung cancer, hepatocellular carcinoma, and colorectal cancer. (Adapted from Kaplan–Meier plotter: https://kmplot.com/analysis/index.php?p=home (accessed on 1 November 2025).

**Figure 5 pharmaceuticals-18-01784-f005:**
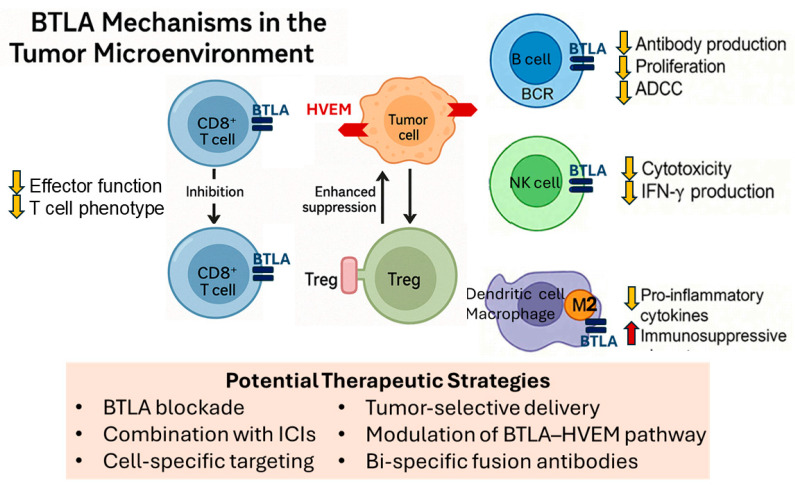
BTLA mechanisms in the tumor microenvironment. Schematic illustration showing BTLA expression on multiple immune cell types and its immunosuppressive effects. BTLA expressed on CD8^+^ TILs engages HVEM on tumor cells, leading to downstream inhibitory signaling. This results in reduced effector function, cytokine production, and the induction of an exhausted T cell phenotype, thereby limiting cytotoxic responses. BTLA is also expressed on Tregs, where it contributes to the stabilization and enhancement of their suppressive phenotype. HVEM engagement promotes increased Treg-mediated inhibition of local immune activity, reinforcing immune tolerance at the tumor site. Together, these mechanisms create a bidirectional immunosuppressive axis that facilitates tumor immune evasion and limits the efficacy of immunotherapy. In B cells, BTLA engagement reduces antibody production, proliferation, and ADCC. In NK cells, BTLA signaling decreases cytotoxic activity and IFN-γ production. In dendritic cells and macrophages, BTLA promotes tolerogenic and M2-like phenotypes by suppressing pro-inflammatory cytokines and enhancing immunosuppressive properties. Potential therapeutic strategies include BTLA blockade alone or in combination with ICIs, cell-specific targeting, tumor-selective delivery, modulation of the BTLA–HVEM pathway, or bi-specific fusion antibodies to overcome multiple layers of immune suppression. (Abbreviations: BTLA: B and T lymphocyte attenuator; TIL: tumor-infiltrating lymphocyte; HVEM: herpesvirus entry mediator; Treg: regulatory T cell; ADCC: antibody-dependent cell-mediated cytotoxicity; NK cell: natural killer cell; IFN-γ: interferon-gamma; ICIs: immune checkpoint inhibitors).

**Figure 6 pharmaceuticals-18-01784-f006:**
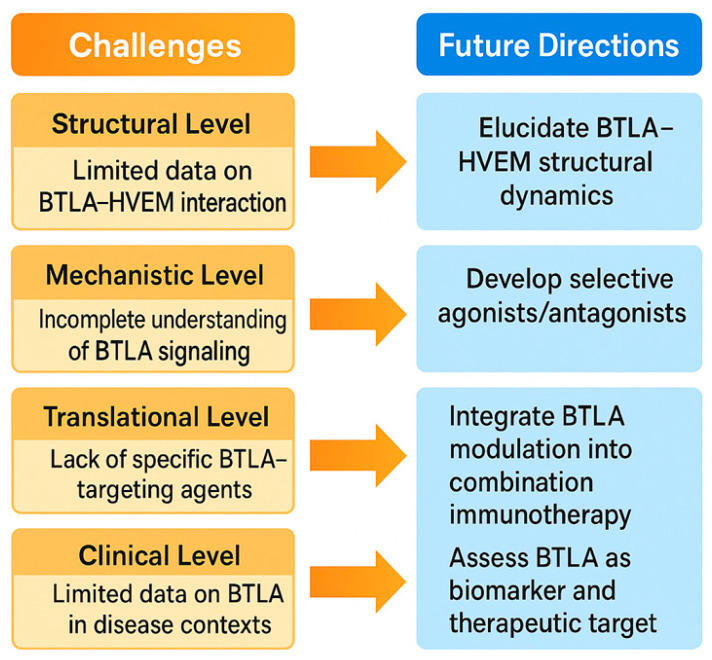
This schematic illustrates the key challenges and future directions in BTLA-targeted immunotherapy. On the left, major current limitations are depicted, including heterogeneous BTLA expression across tumors, dual functional roles of the BTLA–HVEM pathway, limited clinical efficacy in certain cancer types, and safety concerns related to immune overactivation. A critical unmet need is the identification of reliable predictive biomarkers—such as BTLA/HVEM expression profiles, immune gene signatures, and functional immune assays—to guide patient selection. On the right, innovation opportunities and personalized medicine strategies are presented. These include the development of next-generation BTLA modulators such as bispecific antibodies, Fc-engineered molecules, small-molecule inhibitors, and agonists for specific immune contexts. The integration of multi-omics data, AI-driven predictive modeling, and adaptive clinical trial designs offers a path toward tailoring BTLA-targeted therapies to individual tumor immune landscapes, aiming to maximize efficacy while minimizing adverse effects.

**Table 1 pharmaceuticals-18-01784-t001:** Summary of BTLA expression trends and their prognostic significance across different types of cancer.

Cancer Type	BTLA Expression Pattern	Clinical Prognostic Significance	References
Non-small cell lung cancer (NSCLC)	Highly expressed in TILs; associated with immunosuppression and advanced disease stage	Reduced overall survival (OS) and progression-free survival (PFS)	[31,49,59]
Hepatocellular carcinoma (HCC)	Correlates with vascular invasion and higher tumor grade	Associated with shorter overall and disease-free survival (DFS)	[5,51,60,61]
Melanoma	Linked to immunotherapy resistance and disease relapse	High expressers respond poorly to PD-1 monotherapy	[12,18,44,52,62,63,64,65,66]
Colorectal cancer (CRC)	Upregulated in MSS subtype; indicative of immune exclusion phenotype	Correlates with poor immunotherapy response and lower OS	[53,67,68]
Diffuse large B cell lymphoma (DLBCL)	Expressed on T and B cells; associated with poor-risk subtypes and lower survival	Associated with inferior event-free survival (EFS)	[54,69,70]
Chronic lymphocytic leukemia (CLL)	Co-expressed with exhaustion markers; linked to treatment resistance	Linked to poor response to immunotherapy and disease progression	[4,46,71]
Epithelial ovarian cancer (EOC)	Detected in cancerous tissues and plasma of EOC patients	High expression levels correlate with poor outcomes	[55,56,57]

**Table 2 pharmaceuticals-18-01784-t002:** Therapeutic agents targeting the BTLA/HVEM axis.

Agent	Molecular Type	Mechanism of Action	Development	Target Indications	Refs.
Tifcemalimab (JS004/TAB004, Junshi Biosciences, Shanghai, China)	Recombinant humanized IgG4κ mAb	BTLA antagonist- blocks the BTLA/HVEM interaction to restore T cell activation	Phase 2/3 ongoing in solid tumors and lymphoma (mono- and combo-therapy with toripalimab)	Relapsed/refractory lymphoma, NSCLC, SCLC, ESCC, solid tumors, autoimmune exploration	[59,89,90,91]
ANB032 (AnaptysBio; San Diego, CA, USA)	Human IgG4 non-depleting mA	BTLA agonist- enhances BTLA inhibitory signaling to suppress inflammation without blocking HVEM	Phase 2b (AD trial completed; did not meet primary endpoint)	Atopic dermatitis, inflammatory and autoimmune diseases	[92,93]
LY3361237 (Venanprubart, Eli Lilly and Company, Indianapolis, IN, USA)	Human IgG4 mAb	BTLA agonist-activates BTLA to down-modulate autoreactive T/B cells	Phase 2 in progress (SLE and primary Sjögren’s syndrome)	Systemic lupus erythematosus, primary Sjögren’s syndrome	[94]
HFB200603 (HiFiBiO Therapeutics, Cambridge, MA, USA)	Humanized IgG1 mAb	BTLA antagonist- blocks BTLA/HVEM signaling to relieve tumor-induced immunosuppression	Phase 1 (monotherapy ± tislelizumab; ESMO 2024 update)	Solid tumors, immuno-oncology combinations	[95,96]

**Table 3 pharmaceuticals-18-01784-t003:** Clinical development landscape of BTLA-targeted immunotherapies.

Agent	Phase/NCT	Indication and Population	Study Design and Regimen	Key Efficacy Findings	Ref.
Tifcemalimab	Phase I (NCT04477772)	Relapsed/refractory lymphoma (esp. cHL post-PD-(L)1)	Dose escalation and expansion; monotherapy and in combination with toripalimab (anti-PD-1)	cHL cohort: ORR 37%, median PFS 13.1 mo; durable responses in PD-(L)1-refractory cases	[91]
Tifcemalimab + Toripalimab	Phase I/II (NCT05000684)	Advanced or previously treated NSCLC/SCLC	Multi-cohort combination immunotherapy	NSCLC: ORR 4.3%, DCR 47.8%, mPFS 1.5 mo, mOS 18.9 mo; SCLC: ORR 35%, DCR 55%, mPFS 2.8 mo, mOS 12.3 mo	[59]
Tifcemalimab + Toripalimab	Phase I/II (NCT05000684)	Refractory extensive-stage SCLC	Combination therapy after prior systemic therapy	Reported promising antitumor activity (ORR 26.3%, DCR 57.9%)	[59]
Tifcemalimab + Toripalimab + Docetaxel	Phase Ib/II	Second-line squamous NSCLC after immunotherapy	Triple combination (BTLA+ PD-1+ chemotherapy)	Early data: 6- and 9-month OS rates 85.3%, 70.1%	[89]
Peri-operative ESCC (BT-NICE) -tifcemalimab + toripalimab ± chemo	Phase II (NCT06588335)	Resectable esophageal squamous cell carcinoma	Neoadjuvant tifcemalimab + toripalimab + chemo → surgery; adjuvant tifcemalimab + toripalimab (± chemo/RT)	Ongoing; endpoints: pCR rate, DFS, OS	[90]
Tifcemalimab + Toripalimab (Consolidation)	Phase III (NCT06095583)	Limited-stage SCLC post-concurrent chemoradiotherapy, no progression	Double-blind, placebo-controlled, 3-arm trial: toripalimab ± tifcemalimab vs. placebo	Ongoing; evaluating PFS/OS	[98]
Tifcemalimab + Toripalimab ± Chemotherapy	Phase II (NCT05664971)	Advanced or metastatic NSCLC previously treated with immunotherapy ± chemotherapy	tifcemalimab + toripalimab ± platinum-based chemotherapy	Ongoing; primary endpoints: ORR and safety	[99]
Tifcemalimab + Toripalimab	Phase II (NCT06648200)	Limited-stage SCLC following concurrent chemoradiotherapy	Prospective, multicenter trial, as consolidation therapy post-CRT	Ongoing; endpoints: PFS/OS; designed to precede the confirmatory Phase III trial (NCT06095583)	[100]
Tifcemalimab + Toripalimab ± Chemo/Radiotherapy	Phase II (NCT06732258)	Advanced or locally advanced solid tumors (exploratory)	Investigator-initiated, open-label basket trial testing BTLA + PD-1 blockade ± standard chemo/radiotherapy	Recruiting; no efficacy data yet; aims to identify tumor types most responsive to dual blockade	[101]
Tifcemalimab + Toripalimab	Phase II (NCT06690697)	Clear-cell renal cell carcinoma	Randomized single-center trial examining combined therapy vs. control	Ongoing (no efficacy data yet)	[102]
HFB200603 (HiFiBiO)	Phase I (NCT05789069)	Advanced solid tumors	Monotherapy and in combination with tislelizumab (PD-1)	Early poster: disease stabilization and preliminary responses across doses	[96]

cHL: classical Hodgkin lymphoma; NSCLC: non-small cell lung cancer; SCLC: small cell lung cancer.

## Data Availability

Not applicable.

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
