# Peer review of "BTLA: An Emerging Immune Checkpoint Target in Cancer Immunotherapy"

_pharmaceuticals, 2025, doi:10.3390/ph18121784_

Round 1
Reviewer 1 Report
Comments and Suggestions for Authors
- The section on BTLA highlights its role as an emerging immune checkpoint; however, the discussion would be strengthened by incorporating clinical evidence. Several recent studies, including correlative analyses in NSCLC, AML, and HCC, as well as early-phase clinical trials evaluating anti-BTLA antibodies (e.g., icatolimab, tifcemalimab), provide important translational and clinical context. Including these references would improve the comprehensiveness of the review
- For Table 1, please add supporting references for each entry to strengthen the data and improve credibility
- The quality of Figure 3 is not satisfactory. It would be better to revise the presentation style and adopt a more consistent figure format.
- The quality of the tables and figures is not satisfactory. In a review article, figures and tables should comprehensively highlight key concepts and important information; however, in the present manuscript they do not adequately achieve this.
Author Response
Comments 1: The section on BTLA highlights its role as an emerging immune checkpoint; however, the discussion would be strengthened by incorporating clinical evidence. Several recent studies, including correlative analyses in NSCLC, AML, and HCC, as well as early-phase clinical trials evaluating anti-BTLA antibodies (e.g., icatolimab, tifcemalimab), provide important translational and clinical context. Including these references would improve the comprehensiveness of the review
Response 1: Thanks for the comment and we really appreciate it. We have revised it in Table 2 which includes recent clinical and translational findings on BTLA (Please see Table 2).
Comments 2: For Table 1, please add supporting references for each entry to strengthen the data and improve credibility
Response 2: Thanks for the comment and we really appreciate it. We have revised it in Table 1 (Please see Table 1).
Comments 3: The quality of Figure 3 is not satisfactory. It would be better to revise the presentation style and adopt a more consistent figure format.
Response 3: Thanks for the comment and we really appreciate it. We have revised it in Figure 6 (Please see Figure 6).
Comments 4: The quality of the tables and figures is not satisfactory. In a review article, figures and tables should comprehensively highlight key concepts and important information; however, in the present manuscript they do not adequately achieve this.
Response 4: Thanks for the comment and we really appreciate it. We have revised it (Please see Tables and Figures).
Reviewer 2 Report
Comments and Suggestions for Authors
The authors review the role of BTLA in cancer Immunotherapy
This review is very comprehensive covering all aspects of the BTLA-HVEM (TNFRSF14) axis.
General comments
Unfortunately, it is also very repetitive. Many things such as cell-specific expression are repeated in many paragraphs across all chapters e.g. lines 59, 77, 91. There are many other things constantly repeating. Please, make this review much more concise and avoid repeating the same facts in each chapter again. Making this review as concise as possible could also be done by decreasing the number of chapters and focusing on the main aspect for each chapter. An example for this is chapter 5.2. (Preclinical and clinical studies investigating BTLA inhibition.): Many other clinical trials and preclinical studies are also mentioned in chapter 5.3, 6.1 and other chapters.
The review could be improved tremendously by adding additional figures (schematics) and tables.
Please add a table with all clinical trials with references.
Please combine the BTLA/HVEM mechanism of figure 1 (T cells) with figure 2 (B, NK and myeloid cells), that would make a more comprehensive figure.
What are the BTLA/HVEM expression levels among cell types? Maybe a figure would illustrate that using data from public sources such as The Human Protein Atlas etc.
Table 1 should be accompanied with Kaplan-Meier Plots for each cancer type (add ovarian cancer)
A table with all BTLA/HVEM targeting agents with references would be helpful accompanying chapter 5.1.
I do not understand the sudden focus on ovarian cancer (missing in table 1) in chapter 5.2 but not covering other cancers mentioned in table 1. Please discuss those cancers in chapter 5.2
Other comments
In figure 2 right panel does “M” stand for “M2” Macrophage? Pro-inflammatory cytokines should have a downwards arrow.
Reference 86 is about a different agent called TAB004: an anti-Muc1 antibody, please correct.
Is there a reference for the statement about clinical trials on line 582 chapter 6.1?
Line 629, are there examples (ref) on how bispecific antibody could be targeting only the TME?
Figure 3, there are more aspects/topics in the legend and in the text than in this figure (which is more of a table). Please add those. Making a schematic would be great to turn it into a real figure.
Line 688, Please add a reference for this statement.
Author Response
General comments
Comments 1: Unfortunately, it is also very repetitive. Many things such as cell-specific expression are repeated in many paragraphs across all chapters e.g. lines 59, 77, 91. There are many other things constantly repeating. Please, make this review much more concise and avoid repeating the same facts in each chapter again. Making this review as concise as possible could also be done by decreasing the number of chapters and focusing on the main aspect for each chapter. An example for this is chapter 5.2. (Preclinical and clinical studies investigating BTLA inhibition.): Many other clinical trials and preclinical studies are also mentioned in chapter 5.3, 6.1 and other chapters.
Response 1: Thanks for the comment and we really appreciate it. We have revised the manuscript to remove redundant descriptions and merged overlapping sections. Several chapters have been reorganized to improve flow and avoid repetition, particularly in Sections 5.2 and 5.3 (Please see Sections 5.2 and 5.3).
Comments 2: The review could be improved tremendously by adding additional figures (schematics) and tables.
Response 2: Thanks for the comment and we really appreciate it. We have revised it (Please see Tables and Figures).
Comments 3: Please add a table with all clinical trials with references.
Response 3: Thanks for the comment and we really appreciate it. We have revised it (Please see Table 2).
Comments 4: Please combine the BTLA/HVEM mechanism of figure 1 (T cells) with figure 2 (B, NK and myeloid cells), that would make a more comprehensive figure.
Response 4: Thanks for the comment and we really appreciate it. We have revised it (Please see Figure 5).
Comments 5: What are the BTLA/HVEM expression levels among cell types? Maybe a figure would illustrate that using data from public sources such as The Human Protein Atlas etc.
Response 5: Thanks for the comment and we really appreciate it. We have revised it (Please see Figure 3).
Comments 6: Table 1 should be accompanied with Kaplan-Meier Plots for each cancer type (add ovarian cancer)
Response 6: Thanks for the comment and we really appreciate it. We have revised it (Please see Figure 4 and Page 7 Line 265-270).
Comments 7: A table with all BTLA/HVEM targeting agents with references would be helpful accompanying chapter 5.1.
Response 7: Thanks for the comment and we really appreciate it. We have revised it (Please see Table 3).
Comments 8: I do not understand the sudden focus on ovarian cancer (missing in table 1) in chapter 5.2 but not covering other cancers mentioned in table 1. Please discuss those cancers in chapter 5.2
Response 8: Thanks for the comment and we really appreciate it. We have revised it (Please see Table 1-3 and chapter 5.2).
Other comments
Comments 9: In figure 2 right panel does “M” stand for “M2” Macrophage? Pro-inflammatory cytokines should have a downwards arrow.
Response 9: Thanks for the comment and we really appreciate it. We have revised it (Please see Figure 5).
Comments 10: Reference 86 is about a different agent called TAB004: an anti-Muc1 antibody, please correct.
Response 10: Thanks for the comment and we really appreciate it. (Please see Table 3 and reference 96).
Comments 11: Is there a reference for the statement about clinical trials on line 582 chapter 6.1?
Response 11: Thanks for the comment and we really appreciate it. We have revised it (Please see Page 17 Line 657 and reference 112).
Comments 12: Line 629, are there examples (ref) on how bispecific antibody could be targeting only the TME?
Response 12: Thanks for the comment and we really appreciate it. We have revised it (Please see Page 19 Line 709 and reference 120-121).
Comments 13: Figure 3, there are more aspects/topics in the legend and in the text than in this figure (which is more of a table). Please add those. Making a schematic would be great to turn it into a real figure.
Response 13: Thanks for the comment and we really appreciate it. We have revised it in Figure 6 (Please see Figure 6).
Comments 14: Line 688, Please add a reference for this statement.
Response 14: Thanks for the comment and we really appreciate it. We have revised it (Please see Page 20 Line 773 and reference 4).
Reviewer 3 Report
Comments and Suggestions for Authors
- The title can still be more concise and clearer.
- The abstract should highlight the unique aspects and therapeutic significance of BTLA.
- Including a figure in the introduction will improve the clarity and understanding.
- The introduction mainly repeats known concepts about immune checkpoints without clearly highlighting the specific importance or novelty of BTLA. The flow can be improved by focusing more on the knowledge gaps and the rationale for reviewing BTLA in detail.
- While talking about PD-L1 inhibitors in the introduction, cite the following relevant article: 10.3390/genes14071370.
- Under sub-heading 2, give a paragraph comparing BTLA with other checkpoints and briefly discuss how its structural features translate into therapeutic potential.
- A figure for “Molecular Structure and Signalling Pathway of BTLA” subsection can enhance the information.
- Under sub-heading 3, highlight which cancer types show the most therapeutic promise.
- Figures 1 and 2 are difficult to interpret. They should be redrawn with clearer labelling, simplified layout, and better spacing to improve readability and visual clarity. Figure 3 can be redesigned to be more visually appealing rather than presented as an information chart.
- Under subheading 5, adding a table summarizing current preclinical and clinical BTLA-targeting agents (drug name, mechanism, trial ID, cancer type, status) would increase translational utility.
- Clarify abbreviations in figure captions (e.g., HVEM, TILs, Tregs).
- The conclusion could more explicitly address translational gaps, such as the limitations of preclinical models in predicting clinical efficacy and the challenges in designing clinical trials that incorporate BTLA-targeted agents.
Author Response
Comments 1: The title can still be more concise and clearer.
Response 1: Thanks for the comment and we really appreciate it. We have revised the title as “BTLA: An Emerging Immune Checkpoint Target in Cancer Immunotherapy”.
Comments 2: The abstract should highlight the unique aspects and therapeutic significance of BTLA.
Response 2: Thanks for the comment and we really appreciate it. In the revised version, we have modified the abstract to better emphasize the unique aspects and therapeutic significance of BTLA. Specifically, we have (1) highlighted BTLA’s distinct dual regulatory roles in immune modulation compared with other immune checkpoints, (2) underscored its emerging importance as a novel therapeutic target in tumors with resistance to PD-1/CTLA-4 blockade, and (3) emphasized its potential for combination immunotherapy and biomarker development (Please see abstract).
Comments 3: Including a figure in the introduction will improve the clarity and understanding.
Response 3: Thanks for the comment and we really appreciate it. We have added Figure 1 (Please see Figure 1).
Comments 4: The introduction mainly repeats known concepts about immune checkpoints without clearly highlighting the specific importance or novelty of BTLA. The flow can be improved by focusing more on the knowledge gaps and the rationale for reviewing BTLA in detail.
Response 4: Thanks for the comment and we really appreciate it. We have revised it (Please see Introduction section).
Comments 5: While talking about PD-L1 inhibitors in the introduction, cite the following relevant article: 10.3390/genes14071370.
Response 5: Thanks for the comment and we really appreciate it. We have revised it (Please see Page 2 Line 64 and reference 10).
Comments 6: Under sub-heading 2, give a paragraph comparing BTLA with other checkpoints and briefly discuss how its structural features translate into therapeutic potential.
Response 6: Thanks for the comment and we really appreciate it. We have revised it (Please see Page 3 Line 82-92).
Comments 7: A figure for “Molecular Structure and Signalling Pathway of BTLA” subsection can enhance the information.
Response 7: Thanks for the comment and we really appreciate it. We have revised it in Figure 2 (Please see Figure 2).
Comments 8: Under sub-heading 3, highlight which cancer types show the most therapeutic promise.
Response 8: Thanks for the comment and we really appreciate it. At present, most available evidence is derived from murine tumor models and in vitro studies, which do not fully capture the complexity of human tumor-immune interactions. Consequently, it remains premature to conclude which cancer types would most benefit from BTLA blockade. Ongoing and future clinical investigations are expected to clarify the tumor contexts in which BTLA inhibition may achieve the highest therapeutic efficacy.
Comments 9: Figures 1 and 2 are difficult to interpret. They should be redrawn with clearer labelling, simplified layout, and better spacing to improve readability and visual clarity. Figure 3 can be redesigned to be more visually appealing rather than presented as an information chart.
Response 9: Thanks for the comment and we really appreciate it. We have revised it in Figure 5 and 6 (Please see Figure 5 and 6).
Comments 10: Under subheading 5, adding a table summarizing current preclinical and clinical BTLA-targeting agents (drug name, mechanism, trial ID, cancer type, status) would increase translational utility.
Response 10: Thanks for the comment and we really appreciate it. We have revised it in Table 2 and 3 (Please see Table 2 and 3).
Comments 11: Clarify abbreviations in figure captions (e.g., HVEM, TILs, Tregs).
Response 11: Thanks for the comment and we really appreciate it. We have revised it in Figure 5 (Please see Figure 5).
Comments 12: The conclusion could more explicitly address translational gaps, such as the limitations of preclinical models in predicting clinical efficacy and the challenges in designing clinical trials that incorporate BTLA-targeted agents.
Response 12: Thanks for the comment and we really appreciate it. We have revised it in conclusion section (Please see Page 21 Line 808-817).
Reviewer 4 Report
Comments and Suggestions for Authors
Manuscript pharmaceuticals-3909204
„BTLA: A Critical Immune Checkpoint in Tumor Immunology and Emerging Target in Cancer Immunotherapy” for Pharmaceuticals
The work is interesting. The authors attempted to describe the basic mechanisms and role of BTLA in cancer immunology. The use of BTLA inhibitors and its use as a marker are of interest.
Comments:
- Please further specify the role of BTLA depending on the stage of tumor development.
- Paragraph 6.2. Please further elaborate on the role of BTLA in the context of macrophages, particularly the M1 to M2 transition.
- Please elaborate on the role of BTLA in angiogenesis and lymphangiogenesis.
- Please briefly describe the role of BTLA expression in metastasis.
Author Response
Reviewer 4
Manuscript pharmaceuticals-3909204
„BTLA: A Critical Immune Checkpoint in Tumor Immunology and Emerging Target in Cancer Immunotherapy” for Pharmaceuticals
The work is interesting. The authors attempted to describe the basic mechanisms and role of BTLA in cancer immunology. The use of BTLA inhibitors and its use as a marker are of interest.
Comments:
Comments 1: Please further specify the role of BTLA depending on the stage of tumor development.
Response 1: Thanks for the comment and we really appreciate it. We have revised it in conclusion section (Please see Page 12 Line 439-446).
Comments 2: Paragraph 6.2. Please further elaborate on the role of BTLA in the context of macrophages, particularly the M1 to M2 transition.
Response 2: Thanks for the comment and we really appreciate it. We have revised it in conclusion section (Please see Page 18 Line 689-697). Although our study mainly focused on lymphocyte-mediated mechanisms, we agree that BTLA’s influence on macrophage polarization represents an important direction for future research, and we have added this discussion in the revised manuscript.
Comments 3: Please elaborate on the role of BTLA in angiogenesis and lymphangiogenesis.
Ans: We appreciate the reviewer’s insightful comment.
Response 3: Thanks for the comment and we really appreciate it. BTLA, although primarily known for its immunoregulatory role, may also contribute to angiogenesis and lymphangiogenesis by modulating cytokines such as VEGF, IL-10, and TGF-β within the tumor microenvironment. This immunosuppressive milieu promotes endothelial activation and vessel formation, facilitating tumor progression. We have added a brief illustration on this aspect in the revised manuscript (Please see Page 10 Line 344-353).
Comments 4: Please briefly describe the role of BTLA expression in metastasis.
Response 4: Thanks for the comment and we really appreciate it. Elevated BTLA expression has been associated with enhanced tumor metastasis, mainly through its immunosuppressive effects in the tumor microenvironment. By inhibiting effector T-cell function and promoting the release of cytokines such as IL-10 and TGF-β, BTLA facilitates immune evasion and tumor cell dissemination. This discussion has been added in the revised manuscript (Please see Page 10 Line 344-353).